# Phlorotannins: Novel Orally Administrated Bioactive Compounds That Induce Mitochondrial Dysfunction and Oxidative Stress in Cancer

**DOI:** 10.3390/antiox12091734

**Published:** 2023-09-07

**Authors:** Layla Simón, Migdalia Arazo-Rusindo, Andrew F. G. Quest, María Salomé Mariotti-Celis

**Affiliations:** 1Nutrition and Dietetic School, Facultad de Medicina, Universidad Finis Terrae, Santiago 7501015, Chile; 2Cellular Communication Laboratory, Center for Studies on Exercise, Metabolism and Cancer (CEMC), Program of Cell and Molecular Biology, Institute of Biomedical Sciences (ICBM), Faculty of Medicine, Universidad de Chile, Santiago 8380000, Chile; aquest@u.uchile.cl; 3Department of Chemical and Bioprocess Engineering, Faculty of Engineering, Pontificia Universidad Católica de Chile, Santiago 7820436, Chile; migdalia.arazo@uc.cl; 4Advanced Center for Chronic Diseases (ACCDiS), Faculty of Medicine, Universidad de Chile, Santiago 8380000, Chile

**Keywords:** metabolism, antioxidants, pro-oxidants, phloroglucinol, encapsulation

## Abstract

Mitochondrial dysfunction is an interesting therapeutic target to help reduce cancer deaths, and the use of bioactive compounds has emerged as a novel and safe approach to solve this problem. Here, we discuss the information available related to phlorotannins, a type of polyphenol present in brown seaweeds that reportedly functions as antioxidants/pro-oxidants and anti-inflammatory and anti-tumorigenic agents. Specifically, available evidence indicates that dieckol and phloroglucinol promote mitochondrial membrane depolarization and mitochondria-dependent apoptosis. Phlorotannins also reduce pro-tumorigenic, -inflammatory, and -angiogenic signaling mechanisms involving RAS/MAPK/ERK, PI3K/Akt/mTOR, NF-κB, and VEGF. In doing so, they inhibit pathways that favor cancer development and progression. Unfortunately, these compounds are rather labile and, therefore, this review also summarizes approaches permitting the encapsulation of bioactive compounds, like phlorotannins, and their subsequent oral administration as novel and non-invasive therapeutic alternatives for cancer treatment.

## 1. Introduction

Cancer cells acquire capabilities that enable tumor growth and metastatic dissemination. Metabolic reprogramming is one of the hallmarks of cancer. The increased cell proliferation observed in neoplastic disease is associated with adjustments in the energy metabolism in order to permit accelerated cell growth and division. Cancer cells metabolize glucose to lactate instead of processing the molecule via mitochondrial respiration [1,2]. Although less efficient, this allows cancer cells to obtain energy, as well as intermediate metabolites required to proliferate and metastasize. Specifically, glycolytic metabolites fuel the biosynthesis of nucleosides and amino acids, essential for the synthesis of macromolecules required during proliferation. Moreover, glycolysis has been associated with the activation of signaling pathways involving proteins such as RAS (rat sarcoma virus oncogene), Myc (Myc proto-oncogene), TP53 (tumor protein suppressor p53), and HIF-1α (hypoxia inducible factor 1 subunit alpha), thereby promoting cancer development and progression [1,3,4,5,6,7,8,9,10,11]. Furthermore, cancer cells display mitochondrial DNA mutations which induce mitochondrial dysfunction associated with tumor development, progression, and chemo-resistance. For instance, mutations in mitochondrial NADH dehydrogenase result in reduced activity of the mitochondrial respiratory Complex I and increased formation of reactive oxygen species (ROS), which increase the metastatic potential of murine Lewis lung carcinoma cells [12]. In addition, some mitochondrial DNA mutations increase ROS levels, resulting in increased Akt (also known as protein kinase B, PKB), MAPK (mitogen-activated protein kinase), and HIF-1α-dependent signaling pathways, thereby promoting cancer progression [13,14,15]. Meanwhile, partial depletion of mitochondrial DNA increases anti-apoptotic Bcl-2 (B-cell lymphoma 2) and Bcl-X(L) (B-cell lymphoma-extra-large) protein levels and induces the sequestration of the proapoptotic factors Bid (BH3-interacting domain death agonist), Bax (Bcl-2-like protein 4) and Bad (Bcl-2 associated agonist of cell death) in the inner mitochondrial membrane, thus preventing cell death through apoptosis, which again favors cancer development [16,17].

ROS are generated as byproducts of the mitochondrial electron transport chain and NADPH oxidases. In addition, peroxisomes and endoplasmic reticulum (ER) membranes generate ROS. Mitochondrial ROS are mainly produced by complexes I and III, and while 50% is retained within the mitochondrial matrix, the other 50% is released to the cytoplasm [18]. Subsequently, ROS inhibit the phosphatases PTEN (phosphatase and tensin homolog) and PTP1B (protein tyrosine phosphatase non-receptor type 1), negative regulators of the PI3K/Akt/mTOR and MAPK/ERK mitogenic signaling cascades, thereby driving the survival and proliferation of cancer cells [19,20,21]. Furthermore, ROS promote the epithelial-to-mesenchymal (EMT) transition of cancer cells through cytoskeleton rearrangement induced by Rac1 (Rac family small GTPase 1), RhoA (Ras homolog family member A) and FAK (focal adhesion kinase) [22,23,24]. Moreover, ROS induce NF-κB (nuclear factor kappa-light-chain-enhancer of activated B cells) phosphorylation, increase matrix metalloproteinase (MMP) expression, and enhance extracellular matrix degradation [25,26]. Also, ROS suppress HIF-1α degradation and induce angiogenesis [27,28,29]. Therefore, ROS increase the metastatic potential of cancer cells. On the other hand, excessive ROS increments have antitumoral effects by inducing cancer cell death [27,30,31,32,33]. In this sense, cancer treatment could be possible by reducing or increasing ROS levels, in order to prevent early neoplasia or to kill cancer cells, respectively [27].

Phytochemicals are secondary plant metabolites with antioxidant properties that play important roles in cancer chemoprevention by reversing oxidative stress-induced malignant transformation. Indeed, populations that consume high levels of plant-derived foods enriched in polyphenols have reduced cancer incidence [34]. In this way, curcumin, epigallocatechin gallate, and resveratrol inhibit cancer cell proliferation, survival, migration, invasion and thereby tumor growth and metastasis [27,35]. Hence, utilizing polyphenols as an orally administered therapeutic alternative represents a less invasive approach [36] to cancer prevention.

Phlorotannins are phenolic compounds produced by sea algae with elevated antioxidant capacity compared to polyphenols from terrestrial plants. They have demonstrated bioactive properties, including the reduction of oxidative stress, inflammation and tumorigenesis. Due to the potential of phlorotannins to reduce cancer development and progression, there is great interest in their biopharmaceutical application [34].

Although phlorotannins possess numerous relevant bioactivities, their therapeutic applications, when orally administrated, are limited due to low bioavailability. The primary factors that affect their bioavailability include low solubility, poor stability, low absorption in the human gastrointestinal tract, extensive biotransformation within the gut, and rapid clearance from the body [37]. In this respect, numerous strategies have been employed to enhance the bioavailability of polyphenols, which seek to overcome their limited absorption and utilization in the body. One approach involves the use of food matrices or delivery systems that can improve the solubility and stability of polyphenols during digestion [36].

Encapsulation techniques, such as microencapsulation, nanoemulsions or nanoparticles and liposomes, among others, have shown promise in enhancing polyphenol bioavailability by protecting them from degradation and promoting their absorption [36]. These strategies are being actively investigated and hold considerable potential for maximizing the therapeutic benefits of polyphenols, including phlorotannins, when administered via oral delivery systems in various health applications.

This review will summarize the available literature related to mitochondrial dysfunction-promoting cancer. In addition, we will discuss the evidence available relating to phlorotannins as antitumoral therapeutic agents targeting mitochondrial dysfunction. Finally, we will highlight the rational design of oral delivery systems for phlorotannins using encapsulation techniques as a novel and non-invasive therapeutic alternative in cancer treatment.

## 2. Mitochondrial Dysfunction and Oxidative Stress in Cancer

Cancer cells exhibit mitochondrial dysfunction due to defects in tricarboxylic acid (TCA) cycle enzymes and the mitochondrial electron transport chain, mitochondrial DNA mutations, oxidative stress, as well as aberrant oncogene and tumor suppressor signaling [38,39,40,41,42,43,44,45,46]. Subsequently, mitochondrial dysfunction can promote cancer progression to an apoptosis-resistant/chemo-resistant and/or invasive phenotype through various mechanisms involving KRAS, c-Myc, MAPK, AMPK (AMP-activated protein kinase), PI3K/Akt, HIF-1α and TP53 [15,38,47,48,49,50]. Furthermore, oxidative stress, generated as a consequence of the aberrant mitochondrial metabolism, plays a dual role in normal and cancer cells. For instance, the accumulation of ROS is detrimental in normal cells, but cancer cells maintain high levels of metabolism and generate ROS that facilitate the activation of several signaling pathways and promote cancer progression. However, the increase in ROS beyond a certain threshold level becomes toxic and promotes cancer cell death. For that reason, metabolic tumor reprogramming favors glycolysis and mechanisms transforming pyruvate to lactate, which consumes and reduces ROS to non-toxic levels in cancer cells (reviewed in [51]). 

## 3. Phlorotannins, a Group of Bioactive Compounds with Cancer-Preventing Potential

The ROS balance is highly relevant in cancer therapy. Some chemotherapies increase ROS to toxic levels, thereby promoting cancer cell death. Alternatively, antioxidants that reduce ROS content also serve to prevent signaling pathways related to cancer progression (reviewed in [51]). The metabolic reprogramming from oxidative phosphorylation to a glycolytic metabolism results in cells that generate fewer ROS. Consequently, glycolytic cancer cells are resistant to chemotherapeutic agents that rely on the production of ROS and induction of apoptosis [52]. Furthermore, the level of glycolysis correlates with tumor migration, invasion, and metastasis [2], making the targeting of glycolysis, mitochondria and ROS important approaches for the development of novel therapies.

Polyphenols are bioactive compounds widely present in terrestrial and marine plants that display antioxidant and anti-inflammatory properties [53,54,55]. In addition, they have been attributed anti-tumorigenic properties, whereby some bioactive compounds prevent cancer through metabolic control. For instance, novel polyphenols inhibit cell growth, glycolysis, and mitochondrial respiration in colorectal cancer cells. One proposed mechanism is the activation of AMPK signaling and induction of caspase-dependent apoptosis [56]. In addition, oleuropein, the main bioactive phenolic component present in olive leaves, prevents the aerobic glycolysis exploited by tumor cells. This reduction in activity is attributable to a significant decrease in glucose transporter-1, protein kinase isoform M2, and monocarboxylate transporter-4 expression in melanoma, colon carcinoma, breast cancer, as well as chronic myeloid leukemia cells [57].

Marine polyphenols are bioactive compounds obtained from seaweeds. These polyphenols are grouped as phlorotannins, simple phenolic acids, flavonoids, and bromophenols. Marine polyphenols indeed possess antioxidant capacity and have demonstrated enzyme inhibitory, antimicrobial, antiviral, anticancer, antidiabetic and anti-inflammatory activities [58,59,60,61,62,63,64,65,66]. Phlorotannins are a type of marine polyphenol found exclusively in high levels in brown seaweeds. They are phloroglucinol polymers and exhibit superior antioxidant capacity compared to the other families of phenolic compounds [67]. Phlorotannins are classified according to their chemical structure as fucols, phlorethols, fucophlorethols, fuhalols, carmalols and eckols [58,68]. 

For decades, researchers have extracted phlorotannins from brown seaweeds to study their biological properties. Indeed, the ability of phlorotannins to reduce the development and progression of cancer has been evaluated in different models. 

For instance, extracts obtained from *Ecklonia stolonifera* have been probed in hepatocellular cancer cells. There, 100 µM dieckol, a type of phlorotannin enriched in *Ecklonia stolonifera* extracts, increases cytochrome c release and induces apoptosis in Hep3B and Sk-Hep1 liver cancer cells through Bid and caspase-3, 7, 8, and 9-dependent mechanisms (Table 1) [69]. Commercially available dieckol (34 and 67 µM) promotes cell death through the activation of the caspases-3, 8, and 9 in A549 non-small-cell lung cancer cells. Moreover, dieckol reduces migration and invasion of lung cancer cells by decreasing matrix metalloproteinase-9 (MMP-9) and increasing E-cadherin levels. In addition, the PI3K/Akt signaling pathway is downregulated by dieckol, thereby affecting lung cancer cell survival, proliferation, and metastatic potential [70]. 

Another protein relevant to the migration and invasion of cancer cells is HIF-1α, which is increased in cancer and hypoxic conditions. In fact, hypoxia promotes HIF-1α expression, ROS generation, migration and invasion of HT29 colon cancer cells [71]. Like other phlorotannins that are known to be antioxidants [72], *Ecklonia cava*-isolated dieckol (34 mM) also reduces hypoxia-induced HIF-1α and ROS levels, as well as increases E-cadherin expression, thereby inhibiting migration and invasion [71]. Moreover, *Ecklonia cava*-isolated dieckol at a lower concentration (34 µM) attenuates ROS-induced increases in MMP-9 levels, as well as FAK and Rac1 activation, thereby preventing the ROS-enhanced migration and invasion of HT1080 fibrosarcoma [73], as well as B16F10 melanoma cells [74].

In a rat model of *N*-nitrosodiethylamine (NDEA)-induced hepatocarcinogenesis, the oral administration of *Ecklonia cava*-isolated dieckol prevents lipid peroxidation, as well as liver cell damage and promotes the enzymatic and non-enzymatic antioxidant defense system, thereby preventing hepatocarcinogenesis in vivo. In this rat model, *Ecklonia cava*-isolated dieckol and the carcinogenic drug NDEA were administrated simultaneously during 15 weeks. In these experiments, 10 and 20 mg/kg body weight of dieckol administrated orally were observed to have less cancer-preventive effects than 40 mg/kg b.w. of dieckol, revealing thereby dose- and concentration-dependent effects in preventing cancer initiation [75]. The mechanisms proposed are the induction of apoptosis via the intrinsic pathway by decreasing Bcl-2 expression and increasing the expression of Bax, favoring cytochrome c release and caspase-9/3 activation. Moreover, dieckol, present in extracts, reduces VEGF levels, thereby inhibiting angiogenesis. In addition, dieckol inhibits the pro-inflammatory transcription factor NF-κB, as well as reducing COX2 (cyclooxygenase 2) levels in a NDEA-induced hepatocarcinogenesis model [76].

Additionally, in vivo anticarcinogenic effects of eckol, another natural phlorotannin derived from marine brown algae, are reportedly linked to their ability to modulate the immune response in mice with sarcoma. In a mouse model of transplanted sarcoma, eckol derived from brown algae increases TUNEL-positive apoptotic cells via caspase-9/3 activation and down-regulates the expression of Bcl-2, Bax, and EGFR (epidermal growth factor receptor), as well as EGFR phosphorylation. In this xenograft-bearing mouse model, eckol was orally administrated as a pre-treatment for 7 days. Then, tumor cells were subcutaneously implanted and eckol was continuously administrated for another 10 days. Following this protocol, 0.25 and 0.50 mg/kg body weight of eckol administrated orally were found to have less cancer-preventive effects than 1 mg/kg b.w. of eckol, thereby revealing dose- and concentration-dependent effects in preventing cancer promotion [77]. 

As mentioned before, cancer cells undergo metabolic reprogramming by increasing glycolysis to replace mitochondrial metabolism. In this way, cancer cells control ROS levels, providing sufficient conditions to activate favorable signaling pathways without inducing apoptosis. A novel phloroglucinol quinone was identified that targets both cancer cells that depend on glycolytic pathways or mitochondrial metabolism, albeit through different mechanisms. On the one hand, 50 µM phloroglucinol quinone induces mitochondrial membrane depolarization and loss of mitochondria electron transport, thereby reducing ATP synthesis and affecting HL-60 and HeLa cancer cells that maintain basal levels of mitochondrial metabolism. On the other hand, phloroglucinol prevents autophagy and reduces nutrient recycling, thereby affecting glycolytic cancer cells [52].

Other synthetic phloroglucinols, hyperforin and myrtucommulone A acylphloroglucinols, have been shown to reduce HL-60 leukemia cell viability by directly affecting mitochondria at 0.03–0.9 µM. These phloroglucinols act as protonophores that dissipate the mitochondrial membrane potential, eliminate the mitochondrial proton motive force, and reduce ATP synthesis, which in turn activates AMPK and the intrinsic apoptosis pathway [78]. These effects were reviewed previously, showing that phloroglucinol has the ability to prevent cancer development and progression through the inhibition of mitochondrial metabolism and ROS production as well as inflammatory, angiogenic, and metastatic pathways [79,80,81,82].

*Fucus vesiculosus*-derived phlorotannin extracts exert specific cytotoxicity against Caco-2 and HT29 colon and MKN-28 gastric tumor cells without affecting the viability of HFF-1 normal cells. Specifically, eckstolonol and fucofurodiphlorethol, between 0.4 and 2.4 mM phloroglucinol equivalent, induce cell cycle arrest and apoptosis in cancer cells [83]. In addition, 1% phlorotannin-rich extracts derived from the brown algae *Ascophyllum nodosum* and *Fucus vesiculosus* have been shown to reduce ROS levels, thereby preventing cancer progression in A549 lung cancer cells [84].

On the other hand, phlorotannins have also been described as bioactive compounds that prevent cancer progression by increasing oxidative stress. Commercially available dieckol induces ROS generation and apoptosis in MG-63 human osteosarcoma cells through the activation of caspase-3 and the inhibition of the PI3K/Akt/mTOR signaling pathway. Also, 15 µM of commercially available dieckol reduces metalloproteinase levels and inflammatory markers (TNF (tumor necrosis factor), NF-κB, COX2, and IL-6 (interleukine-6)) [85], indicative of anti-metastatic and anti-inflammatory effects.

Moreover, 400 µg/mL *Ecklonia maxima* and *Ulva rigida* extracts induce apoptosis in HepG2 liver cancer cells through inhibition of the mitochondrial membrane potential and increasing ROS [86]. Also, *Ecklonia cava* extracts enriched in dieckol have cytotoxic effects in ovarian cancer cells and reduce tumor growth in xenograft mouse models when administrated orally at 100 mg/kg b.w. during 4 weeks. In vitro, 120 µM dieckol isolated from *Ecklonia cava* induces apoptosis in SKOV3 ovarian cancer cells through the activation of caspase-8,-9 and -3, as well as by inhibiting the Akt signaling pathway. At the mitochondrial level, dieckol induces mitochondrial membrane depolarization and cytochrome c release, as well as increases the expression of pro-apoptotic proteins. Additionally, dieckol downregulates the expression of anti-apoptotic proteins, such as XIAP (X-linked inhibitor of apoptosis protein), FLIP ((FADD-like IL-1β-converting enzyme)-inhibitor protein), and Bcl-2, thereby promoting apoptosis. These effects are associated with ROS increments. Indeed, the antioxidant *N*-acetyl-L-cysteine prevents caspase activation, cytochrome c release, Bcl-2 downregulation, and apoptosis that are caused by exposure to the seaweed extract [87].

Additionally, phlorotannins also improve the effects of chemotherapy by promoting apoptosis in cancer cells while protecting normal cells. Indeed, the combined administration of phlorotannin (dieckol)-rich extracts of *Ecklonia cava* and cisplatin potentiates the effects of each drug administrated alone by inducing apoptosis via the increase in ROS and inhibition of the Akt/NF-κB pathway. Moreover, dieckol-rich extracts suppress cisplatin-induced normal kidney cell damage [88].

**Table 1 antioxidants-12-01734-t001:** Phlorotannins preventing cancer development and progression.

Bioactive Compound	Model Dosis	REDOX Balance	Mechanism	Disease	Reference
Dieckol, isolated from *Ecklonia stolonifera*	Hepatocellular carcinoma (Hep3B and Sk-Hep1) cells 100 µM, 24 h	Antioxidant	Release of cytochrome c from mitochondria and induction of apoptosis	Hepatocellular carcinoma	[69]
Dieckol	Non-small-cell lung carcinoma A549 cell line 25 and 50 µg/mL (34 and 67 µM), 24 h	Antioxidant	Induction of apoptosis through caspases-3, 8, and 9. Inhibition of proliferation and migration by regulating the PI3K/AKT signaling pathway	Non-small-cell lung cancer	[70]
Dieckol	HT29 cells 25 mg/mL (34 mM), 3 and 12 h	Antioxidant	Inhibition of HIF-1α, ROS, migration, and invasion	Colon cancer	[71]
Dieckol, isolated from *Ecklonia cava*	HT1080 cells 25 µg/mL (34 µM), 24 and 48 h	Antioxidant	Reduction of ROS, Rac1, FAK, adhesion, migration, and invasion	Fibrosarcoma	[73]
Dieckol, isolated from *Ecklonia cava*	B16F10 cells 25 µg/mL (34 µM), 24 and 48 h	Antioxidant	Reduction of NADPH oxidase, ROS, Rac1, migration, and invasion	Melanoma	[74]
Dieckol, isolated from *Ecklonia cava*	Rat model of NDEA-induced hepatocarcinogenesis 40 mg/kg, 15 weeks, oral	Antioxidant	Increment of antioxidant enzymes, thereby preventing hepatocarcinogenesis in vivo	Hepatocarcinoma	[75]
Dieckol, isolated from *Ecklonia cava*	Rat model of NDEA-induced hepatocarcinogenesis 40 mg/kg, 15 weeks, oral	Antioxidant	Induction of mitochondria-dependent apoptosis: decreased Bcl-2 and increased Bax, cytochrome c release, and caspase-3 activation. Promotion of inflammation and angiogenesis via NF-κB, COX2, and VEGF.	Hepatocarcinoma	[76]
Eckol	Xenograft-bearing mice 1 mg/kg, pretreatment for 7 days + treatment 10 days, oral	Antioxidant	Increased TUNEL-positive apoptotic cells, increased caspase-3 and caspase-9 activation, and reduced expression of Bcl-2, EGFR and EGFR phosphorylation Stimulation of innate and adaptive immune responses	Sarcoma	[77]
Phloroglucinol, quinone PMT7	HL-60, HeLa, K562, and T98G cells 50 µM, 90 min, 24, 48 and 72 h	Antioxidant	Mitochondrial membrane depolarization and inhibition of autophagy	Leukemia, cervical carcinoma, and glioblastoma	[52]
Acylphloroglucinols, hyperforin, and myrtucommulone A	HL-60 cells EC50 0.03–0.9 µM	Antioxidant	Mitochondrial membrane depolarization and induction of apoptosis	Leukemia	[78]
Eckstolonol and fucofurodiphlorethol, derived from *Fucus vesiculosus*	Caco-2, HT29, MKN-28, and HFF-1 cells 50–300 µg/mL (0.4–2.4 mM phloroglucinol equivalent), 48 h	Antioxidant	Induction of cell cycle arrest and apoptosis	Colon and gastric cancer	[83]
Phlorotannin-rich extract from *Ascophyllum nodosum* and *Fucus vesiculosus*	A549 cells 1% extracts, 20 min preincubation	Antioxidant	Reduction of ROS	Lung cancer	[84]
Commercial dieckol	MG-63 cells 15/20 µM, 24 h	Pro-oxidant	Induction of ROS generation and apoptosis Inhibition of PI3K/AKT/mTOR pathway Reduction of TNF-α, NF-κB, COX2, IL-6, and matrix metalloproteinase levels	Sarcoma	[85]
Extracts from *Ecklonia maxima* and *Ulva rigida*	HepG2 cells 200 and 400 µg/mL extracts, 48 h	Pro-oxidant	Reduction of mitochondrial membrane potential. Increment of ROS. Induction of apoptosis.	Liver cancer	[86]
Dieckol isolated from *Ecklonia cava*	SKOV3 cells 60–120 µM, 24 h SKOV3 xenograft mice model 100 mg/kg, 4 weeks, oral	Pro-oxidant	Induction of apoptosis in cancer cells (mitochondrial membrane depolarization, activation of caspases), thereby reducing cancer cell viability and tumor xenograft growth through the increment of ROS	Ovarian cancer	[87]
Dieckol-rich extract of *Ecklonia cava* and cisplatin	SKOV3 cells 100 µg/mL (800 µM), 24 h SKOV3 xenograft mouse model 100 mg/kg, 4 weeks, oral	Pro-oxidant	Induction of ROS and apoptosis. Inhibition of NF-κB and Akt signaling.	Ovarian cancer	[88]

Based on the aforementioned evidence, commercial and seaweed-isolated phlorotannins are effective at preventing cancer development and progression. In vitro results show potential anti-tumorigenic effects in a wide range of concentrations (15 µM–34 mM). Moreover, pre-clinical experiments demonstrate that dieckol isolated from *Ecklonia cava* reduces cancer initiation and development at 40 and 100 mg/kg b.w. in rats and mice, respectively. In addition, the pre-treatment with lower doses of eckol (1 mg/kg b.w.) prevents cancer promotion in xenograft-bearing mice.

## 4. Delivery Strategies Applied to Phlorotannin Encapsulation

Traditional cancer therapies are invasive, often ineffective due to the development of drug resistance, non-targeted delivery, and adverse side effects. These factors have increased the focus on bioactive phytochemicals as potential alternatives for the treatment of cancer. Information provided in the previous section would indicate that phlorotannins have emerged as powerful chemo-preventive and chemo-protective compounds, as they can promote apoptotic cell death both in vitro and in vivo [89]. 

Despite the multiple health benefits, the application of phlorotannins is challenging, specifically when they are administered orally. The biological activity of these polyphenols is limited mainly by their lack of solubility and permeability, as well as their extensive biotransformation by the gut microbiota, which determines their degree of absorption in the human gastrointestinal tract (GIT) [90,91]. Therefore, the effectivity of phlorotannins is determined by their bioaccessibility and bioavailability, since they need to be properly digested and absorbed in order to access the bloodstream and then reach the appropriate target location [37]. For these reasons, several intrinsic and environmental factors affect these molecules (Figure 1). These include long-term storage, manufacturing and transit through the GIT, all of which play significant roles in either enhancing or diminishing the bioactivity of phlorotannins [92]. In fact, in the absence of appropriate protection, phlorotannins are susceptible to transformation and metabolic processing which render them inactive. Thus, in order to benefit from these biomolecules, stabilization of phlorotannins is essential [93].

Various delivery strategies are currently under investigation to enhance the bioaccessibility and bioavailability of phlorotannins. These approaches aim to facilitate the development of novel biopharmaceutical applications involving brown seaweed as a natural resource of phlorotannins [94,95]. 

Encapsulation is the most common technique and consists of “packaging” the core material with functional activity (phlorotannin) into a polymer matrix or coating material (encapsulant) to form a capsule system (oral delivery system) [96]. The main actions of this technology currently focus on

Protecting the phlorotannins from degradation by environmental factors during their storage (light, oxygen, extreme pH, high temperature, etc.) and gastrointestinal passage [93].Avoiding unfavorable interactions between phlorotannins and other components of food, nutraceuticals or pharmaceutical matrices, such as proteins, lipids or complex carbohydrate macromolecules [97].Masking the unpleasant organoleptic properties of the phlorotannin extracts [98].

The different techniques that can be used to elaborate phlorotannin capsule systems can be divided into physical, chemical, and physicochemical methods (Table 2) [99]. The application of physical processes during encapsulation generally involves dehydration or cooling techniques. In the application of chemical methods, the encapsulating matrix components either undergo chemical reactions between themselves or with the bioactive agent. Physicochemical methods combine chemical interactions with physical processes to form the encapsulating matrix. With these systems, it is possible to control the release of orally administered phlorotannins at the target site, as well as modulate their bioaccesibility and therefore their bioavailability (Table 2). The growing industrial interest in using phlorotannins in pharmaceutical applications has increased efforts to improve their bioaccessibility/bioavailability by designing specific delivery platforms. Nevertheless, compared to phenolic compounds from terrestrial plants, research focused on phlorotannin encapsulation from seaweed is still relatively scarce. For many years, drying techniques have been used to encapsulate different compounds due to their low costs, simple operation and flexibility. Two main drying techniques are used for encapsulation: spray-drying and freeze-drying [99]. Spray-drying was used by Cuong et al. [100] and Nkurunziza et al. [101] for powder preparations of phlorotannin extracts from brown algae. Cuong et al. [100] studied the impact of different spray-drying conditions (flow rate, inlet/outlet temperature, polymer/bioactive relation, and pressure) on phlorotannin content and antioxidant activity of a nano phlorotannin powder prepared from the brown algae *Sargassum serratum*. Under optimal conditions, the antioxidant activity of the encapsulated phlorotannin extract reached the highest values for the total antioxidant reducing power and DPPH free radical scavenging (4.347 ± 0.018 g ascorbic acid equivalent/100^–1^ g, 9.390 ± 0.024 g FeSO_4_ equivalent 100^–1^ g, and 70.02 ± 0.26%, respectively). Different powders of encapsulated brown algae extract from *Saccharina japonica* were prepared through spray-drying encapsulation with different encapsulant polymers (dextrin, maltodextrin, lactose, Arabic gum, whey protein isolate, gelatin, and sodium caseinate). The antioxidant activities (DPPH and ABTS assays) of the powders were higher than for the protein-coated materials compared with when polysaccharides were used [101].

Freeze-drying techniques can improve the stability of phlorotannins by preventing their degradation at high temperatures, as in the spray-drying and oven methods. Anwar et al. [102] prepared a powder of brown seaweed (*Sargassum plagyophyllum*) extract with a maltodextrin dextrose equivalent 10–15 units using a freeze-drying method. The powder formulations significantly improved the stability of phlorotannins during the drying process, but the IC50 (DPPH assay) could not be calculated since the percentage inhibition did not reach 50%. 

Lipid-based delivery systems, such as emulsions/nanoemulsions, solid lipid nanoparticles, nanostructured lipid carriers and liposomes are the techniques widely used to increase the solubility, stability, bioaccessibility and bioavailability of various phenolic compounds [103]. Especially liposomes are well-suited for increasing the stability of phenolic compounds by protecting them from extreme pH, temperatures, and ion concentrations [104].

Liposomes are vesicles consisting of single or multiple bilayers composed of phospholipids, which have one hydrophilic head and two hydrophobic fatty acid tails. Consequently, the biphasic character of liposomes allows the entrapment of hydrophilic, lipophilic and amphiphilic molecules [103]. The interactions between phlorotannins and liposome-based systems have been demonstrated to be an effective strategy for cancer treatment thanks to the site-specific delivery of phenolic compounds [105]. The use of folate-conjugated, PEGylated nanoliposomes for encapsulating a polyphenolic-rich extract from *Kappaphycus alvarezii* increases the cellular uptake efficiency of the bioactive compounds, and thus inhibits the growth and induces the apoptosis of human adenoma MCF-7 breast cancer cells [105].

Further, the antioxidant activity (DPPH, ABTS, and FRAP assays) of nanoliposomes for encapsulation of *Sargassum boveanum* extract was evaluated, showing a controlled release of phenolic compounds at different pH values, whereby the burst in release was observed at pH = 3. The antioxidant capacity of nanoliposomes was lower than for free algae extracts, which could be because a large proportion of the phenolic compounds are inside the core of nanoliposomes [106]. Subsequently, the same liposome-based systems were used as functional ingredients for developing a mayonnaise with antioxidant and antimicrobial properties. The different formulations of mayonnaise prepared with the nanoliposomes not only improved the antimicrobial potential of *Sargassum boveanum* extract against *P. aeruginosa*, *E. coli*, and *B. cereus* but also masked the strong taste and color of extracts, as well as increased the oxidative stability of mayonnaise [106].

Aqueous seaweed extracts from *Sargassum incisifolium* were used for the green synthesis of silver and gold nanoparticles, and their potential as antimicrobial and cytotoxic agents was investigated by Mmola et al. [107]. They observed that silver nanoparticles were toxic to Gram-negative bacteria, while the gold nanoparticles lacked such activity. Likewise, the normal human breast cell line MCF-12a was found to be resistant to the silver nanoparticles, while the colon cancer cell line HT-29 was more sensitive (10% viability). The gold nanoparticles displayed negligible toxicity. 

Depending on the material used for their fabrication and the preparation technique, these systems can differ in terms of size and morphology. Based on the size, the encapsulated particles could be classified as nanocapsules (<0.2 μm), microcapsules (0.2–5000 μm) or macrocapsules (>5000 μm) [108,109]. Regardless of their size and distribution, particles can have different morphologies depending on the materials (active component and encapsulant) and the technique used in their preparation. According to Yoshizawa [110], the morphology of the particles or capsules can be categorized as single or mono-core, multi or poly-core and matrix type.

**Table 2 antioxidants-12-01734-t002:** Different technologies applied to phlorotannin encapsulation.

Method	Technique	Bioactive Compound	Coating Material	Reference
Physical encapsulation	Spray-drying	Phlorotannin extract (*Sargassum serratum)*	Maltodextrin, glucose, and saccharose	[100]
	Freeze-dried	Phlorotannin extract (*Sargassum plagyophyllum*)	Maltodextrin	[102]
Chemical encapsulation	Complexation	Phlorotannin extracts (*Eisenia bicyclis, Ecklonia cava and Ecklonia kurome)*	Soybean protein isolate	[111]
		Phlorotannin extract (*Laminaria digitata)*	β-casein (random coil) and bovine serum albumin (globular)	[112]
		Phlorotannins (80% purity)	Polyvinylpyrrolidone nanoparticles	[113]
		Phlorotannin extracts (*Undaria pinnatifida*)	Myofibrillar protein from *Scomberomorus niphonius*	[114]
	Emulsion	Phlorotannin extract (*Sargassum fusiforme*)	Propylene glycol and glycerol	[115]
	Liposomes	Phlorotannin extract (*Sargassum boveanum)*	Soybean lecithin and glycerol	[106]
		Phlorotannin extract (*Kappaphycus alvarezii)*	Folic acid-PEG-DSPE conjugate	[105]
Physiochemical encapsulation	Spray-drying/ emulsion	Phlorotannin extract (*Saccharina japonica*)	Polysaccharides (dextrin, maltodextrin, lactose and gum arabic) Proteins (whey protein isolate, gelatin and sodium caseinate)	[101]
	Freeze-dried/ complexation	Phlorotannin extract (*Sargassum incisifolium*)	Gold (III) chloride tryhidrate and silver nitrate (AgNO_3_)	[107]
		Phlorotannin extract (*Ecklonia cava)*	Silver nitrate (AgNO_3_)	[116]
		Phlorotannin extract (*Sargassum ilicifolium*)	Chitosan and tripolyphos-phate (TPP)	[95]
	Electrospinning	Phlorotannin (Zhengzhou Bainafo Bioengineering Co., Ltd., Zhengzhou, Henan, China)	Polyethylene oxide and sodium alginate	[117]
		Phlorotannin (purchased from Zhengzhou Bainafo Bioengineering Co., Ltd., Zhengzhou, Henan, China)	*Momordica charantia* polysaccharide	[118]

## 5. Essential Variables to Modulate the Bioaccessibility and Bioavailability of Phlorotannin Capsule Systems

The design of an effective oral delivery system involving the encapsulation of phlorotannins must consider several variables that play a key role in influencing the interactions between the carriers and the biological interface in the GIT. It is essential to select encapsulants that are suitable for the specific pH conditions in each segment of the GIT and are also compatible with GIT enzymes and gut microbiota. Furthermore, various physicochemical properties of particles, such as size, shape, and surface properties, have been shown to determine the bioaccessibility of phlorotannin capsule systems.

### 5.1. Composition and Chemical Nature of the Encapsulants

The nature and chemical composition of the encapsulants, including polysaccharide/protein complexes or polysaccharide/protein/lipid systems with different concentrations and polymer ratios, can influence phlorotannin bioaccesibility. Together, these inherent characteristics determine the behavior, interactions and performance of the encapsulant during gastrointestinal passage. According to Cassani et al. [93], the complexes formed between polyphenols and carbohydrates or proteins are poorly digested in the upper intestinal tract because of steric hindrance. They reach the colon almost unaltered, where they serve as substrates for the microbial community, resulting in readily absorbable metabolites. O’Sullivan et al. [119] demonstrated that the antioxidant activity of phlorotannin-rich milk samples (DPPH radical scavenging assay) remained stable after a simulated in vitro digestion. It was suggested that the passage of phlorotannins through the GIT should not be affected by interactions with other milk components, such as whey protein isolate, casein and fat. Soybean proteins have also been used to form complexes with phlorotannins extracted from *Eisenia bicyclis*, *Ecklonia cava* and *Ecklonia kurome*. These phlorotannin-soybean protein complexes may be useful as a novel functional foodstuff or supplement with higher DPPH-radical scavenging activity than soybean protein [111].

Moreover, alginate, a natural polysaccharide of brown seaweeds, is resistant to digestion by human enzymes, but can be digested by the human gut microbiota [120]. Conversely, chitosan, a linear polysaccharide derived from chitin, can be partially digested by human enzymes [121]. Thus, through the careful selection of appropriate encapsulants, it becomes feasible to effectively regulate the release of bioactive components from the delivery system precisely at the intended target site.

### 5.2. Physicochemical Properties of Capsule Systems

The physicochemical properties of capsule systems include a range of characteristics, such as size, shape and surface properties. These properties affect various aspects of the encapsulated bioactive compounds, such as their stability, release kinetics and interaction with biological systems [122]. Overall, by carefully considering and optimizing the physicochemical properties of capsules, it is possible to enhance the encapsulation efficiency (EE) of bioactive compounds (EE is defined as the experimental loading/theoretical loading × 100%). This, in turn, leads to more efficient and effective delivery systems. As a reference, effective drug delivery systems usually require an EE to be higher than 60% [37].

The particle size and shape of capsule systems can determine the bio-distribution, cytotoxicity and accumulation of the therapeutic agents in the GIT and/or in the blood stream. Particle size is a critical parameter that affects the behavior of capsule systems because it determines bioavailability, absorption and distribution within the body. Smaller particles tend to result in improved cellular uptake and higher surface area-to-volume ratios, facilitating efficient interactions with target cells or tissues. Furthermore, the shape of the capsule systems can influence their behavior, including their circulation time in the body and interactions with biological barriers [123,124]. 

The surface charges of delivery systems play a significant role in the stability, dispersibility and effectiveness of oral administration [125]. The surface charges are modified using cationic or anionic polysaccharides, such as chitosan or alginate, respectively. Negatively charged particles are trapped by cells of the immune system and diffuse 20 to 30 times faster through the mucous layer than positively charged particles. In contrast, positively charged particles adhere to the mucin of the epithelium [126]. Therefore, a balance between the mucoadhesive and diffusion properties is important for the efficient bioactive release into the mucosa [127]. Consequently, the surface chemistry of the delivery systems defines their biocompatibility, biodistribution, as well as clearance and therefore their bioavailability [124,125].

## 6. Conclusions

In conclusion, cancer occurrence and progression are associated with metabolic reprogramming, increased glycolysis, mitochondrial dysfunction, and ROS production that activate several signaling pathways, thereby promoting proliferation, survival, migration, invasion, and metastasis. On the other hand, phlorotannins reduce cancer development by inhibiting ROS production and the signaling pathways described above. In addition, phlorotannins may prevent cancer through the increment of ROS levels and mitochondrial-dependent apoptosis. In this manner, phlorotannins appear as novel and non-invasive therapeutic alternatives for cancer treatment, but because of their chemical characteristics, their encapsulation for oral administration is required.

## Figures and Tables

**Figure 1 antioxidants-12-01734-f001:**
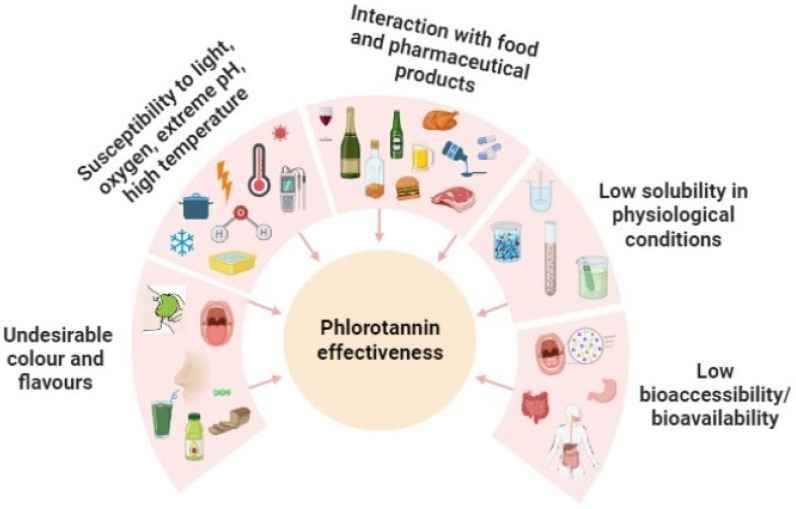
Factors affecting phlorotannins effectiveness as a biopharmaceutical product.

## Data Availability

No new data were created or analyzed in this study. Data sharing is not applicable to this article.

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
