# Peer review of "Phlorotannins: Novel Orally Administrated Bioactive Compounds That Induce Mitochondrial Dysfunction and Oxidative Stress in Cancer"

_antioxidants, 2023, doi:10.3390/antiox12091734_

Round 1

Reviewer 1 Report

Honestly, I can't find what's new in your review. Refer reviews for 35% of your total references. Moreover, some parts of the manuscript look like parts of a teaching textbook. You do not critically review research articles on the topic, or critically point out the shortcomings and challenges that need to be addressed. You use a lot of acronyms without explaining them, making it even more difficult to read.

I enclose the manuscript with my comments in detail.

Reviewer 2 Report

This is a very lengthy review of phlorotannins, oxidative stress and cancer. In general, the review needs to be more focused. I will give some examples but have not listed all the areas that need focus.

1. Lines 17-20 in the abstract could be eliminated. The statements are so general and not helpful. Likewise lines 37-43 are unnecessary for the same reason.

2. Line 108: "This review will summarize the available literature related to mitochondrial dysfunction and oxidative stress-promoting cancer". Any review would not be able to do this. The authors need to concentrate on mitochondrial dysfunction related to phlorotannins.

These examples that are given demonstrate that this review needs to be more focused. 

Round 2

Reviewer 1 Report

You improved your manuscript. You must only present Table 1 somewhere in the text, illustrating it properly.

Reviewer 2 Report

The review is more focused which was a criticism from myself and another reviewer. The authors have removed much  of the unnecessary parts.
